# Effect of Electromagnetic Field on Wear Resistance of Fe901/Al_2_O_3_ Metal Matrix Composite Coating Prepared by Laser Cladding

**DOI:** 10.3390/ma15041531

**Published:** 2022-02-18

**Authors:** Yaobang Chen, Jianzhong Zhou, Pengfei Li, Kun Huo, Xiankai Meng

**Affiliations:** School of Mechanical Engineering, Jiangsu University, Zhenjiang 212013, China; 2211903010@stmail.ujs.edu.cn (Y.C.); 1000005227@ujs.edu.cn (P.L.); 2212103007@stmail.ujs.edu.cn (K.H.); 1000004836@ujs.edu.cn (X.M.)

**Keywords:** laser cladding, electromagnetic field, metal matrix composite coating, wear performance

## Abstract

Fe901/Al_2_O_3_ metal matrix composite (MMC) coatings were deposited on the surface of 45 steel via electromagnetic field (EF)-assisted laser cladding technology. The influences of EF on the microstructure, phase composition, microhardness, and wear resistance of the Fe901/Al_2_O_3_ MMC coating were investigated. The generated Lorentz force (F_L_) and Joule heating due to the application of EF had a positive effect on wear resistance. The results showed that F_L_ broke up the columnar dendrites. Joule heating produced more nuclei, resulting in the formation of fine columnar dendrites, equiaxed dendrites, and cells. The EF affected the content of hard phase in the coatings while it did not change the phase composition of the coating, because the coatings with and without EF assistance contained (Fe, Cr), (Fe, Cr)_7_C_3_, Fe_3_Al, and (Al, Fe)_4_Cr phases. The microhardness under 20 mT increased by 84.5 HV_0.2_ compared to the coating without EF due to the refinement of grains and the increased content of hard phase. Additionally, the main wear mechanism switched from adhesive wear to abrasive wear.

## 1. Introduction

Due to its good comprehensive mechanical properties and low cost, 45 steel has been widely used in the manufacture of various components (such as gears, spindles) [1,2,3]. However, under the action of heavy loads and friction, the component surface is prone to excessive wear failure, which seriously reduces its service life [4]. Therefore, it is necessary to strengthen the surface wear resistance of 45 steel parts to improve the service life of components.

Laser cladding (LC) is an excellent surface modification technology that has been widely used in industry [5,6]. It has the advantage of energy concentration, high efficiency, and small heat-affected zone [7]. Ordinarily, LC can obtain a strong metallurgical bond with the substrate and low dilution rate coatings with dense microstructure [8].

Ceramic particles have the characteristics of high hardness, fatigue resistance, and good toughness, which can effectively improve the surface properties of parts [9,10]. However, it is difficult to prepare pure ceramic coatings because of cracks at the bonding interface during cladding. Therefore, metal matrix composite (MMC) coatings are usually prepared to improve the surface properties of the substrate. Zhu et al. [11] laser-cladded TiC-reinforced 410 martensitic stainless steel (MSS) coatings. The hardness and wear resistance of the coating could be improved by increasing the content of TiC. Wang et al. [12] fabricated a Ni60/Ti_3_SiC_2_ MMC coating on the surface of Ti6Al4V alloy, which greatly improved its surface hardness and wear resistance. However, LC is a very complex physicochemical interaction process. Due to the relatively large differences in thermophysical properties between metal and ceramic particles, defects such as porosity, cracks and uneven composition occur easily in MMC coatings, which greatly affect their preparation and application.

In recent years, an electromagnetic field (EF) has been widely used in casting, welding, and laser processing. Liu et al. [13] used electromagnetic stirring along with a laser to form Inconel 718 superalloy. Electromagnetic stirring led to a phase change from strip-like to sphere-like, while the grain after recrystallization was refined. García-Rentería [14] used electromagnetic interaction of low intensity (EMILI) to assist gas metal arc welding (GMAW). EMILI refined the microstructure and reduced the size of the heat-affected zone. Huo et al. [15] used an electromagnetic compound field (ECF) to assist in laser cladding of IN718/WC coatings. The ECF optimized the uniform distribution of the WC phase, improved the nucleation rate of eutectic carbides, and fractured long columnar dendrites, as well as increased the microhardness by 32.5%.

Previously, our team [16] found that Fe901/Al_2_O_3_ MMC coatings could improve the mechanical performance of a substrate to some extent. When the mass fraction of Al_2_O_3_ was 10%, the microhardness increased by a factor 16.4%, while the average friction factor decreased by 0.85. Moreover, we found that the wear resistance of Fe901/Al_2_O_3_ MMC coatings can be further improved. In this paper, EF-assisted LC was used to prepare Fe901/Al_2_O_3_ MMC coatings with higher wear resistance, and the influence of EF on the microstructural evolution, phase composition, and wear resistance of Fe901/Al_2_O_3_ MMC coatings was explored.

## 2. Materials and Methods

### 2.1. Materials Preparations

In this study, the substrate material was 45 steel (60 mm × 40 mm × 5 mm), and its chemical composition is shown in Table 1. Before the experiment, the surface of the substrate was polished with sandpaper and cleaned with absolute ethanol. Spherical Fe901 powder (size range: 45 to 109 μm) was used in the metal matrix, and its chemical composition is shown in Table 2. Al_2_O_3_ powder (size range: 20 to 40 μm) with an irregular polygonal shape was used to reinforce particles. The Al_2_O_3_ and Fe901 powders were thoroughly mixed with an YXQM 2L planetary ball mill (MITR, China). The mass fraction of Al_2_O_3_ in the composite powder was 10%. The SEM images of the powders are shown in Figure 1. The commercial sources of raw materials are shown in Table 3.

### 2.2. Experimental Procedure

The composite powder was dried at 100 °C for 1 h before LC. The coatings were cladded using a 2 kW continuous wave YLS-2000-TR fiber laser (IPG, New York, NY, USA) with a wavelength of 1070 nm. As shown in Figure 2a, a single coil was placed under the substrate. Figure 2b shows the relationship between the coating and the magnetic induced line during the LC process.

The experimental parameters are shown in Table 4; different electromagnetic field strengths (EFSs) were applied in this study to coatings A, B, and C, where the optimized parameters of laser cladding were selected according to previous experiments [16].

After LC, the sample cross-section was obtained by wire cutting and etched in aqua regia (HCl/HNO_3_ 3:1 *v*/*v*) for 15 s after treatment. The microstructures were observed using a S3400N scanning electron microscope (SEM, HITACHI, Tokyo, Japan). The phases were analyzed using a D8 ADVANCE X-ray diffractometer (XRD, Bruker, Bremen, Germany). The microhardness was measured using an HXD 1000TM Digital Vickers Micro Hardness Tester (TROJAN, Suzhou, China). Twenty points were randomly selected from each LC sample; the average value was adopted, and the standard deviation was calculated.

The width of a single-pass layer was 3 mm, and multiple cladding layers were fabricated for the wear test. As shown in Figure 3a, samples (15 mm × 15 mm × 6 mm) of wear tests were obtained from multiple cladding layers by wire cutting. The height of the samples was 6 mm, including the substrate (5 mm) and coating (1 mm). The samples were polished to achieve a mirror surface. Dry sliding friction and wear tests were carried out on an MFT 5000 linear reciprocating friction and wear tester (RTEC, Wuhan, China) at room temperature. The schematic diagram of volume loss (VL) during the friction and wear process is sketched in Figure 3b. A Si_3_N_4_ ceramic ball was pressed into the LC coating to attain an arc-shaped wear scar section.

A VHX-1000 ultra-depth microscope (Keyence, Itasca, IL, USA) was used to measure the wear depth, width, and 3D morphology of the wear tracks. Each measurement was taken at three points in the wear scar area to obtain an average value. The SEM was used to observe the worn surface.

## 3. Results and Discussion

### 3.1. Microstructure Analysis

During the friction and wear test, the cladding layer needed to be mechanically polished to remove the uneven top area [17]. This paper studied the microstructure at the bottom and middle upper region of the coating. It can be seen in Figure 4(a1–c1) that all of the coatings had bright planar dendrites at the bonding interface. This means that the coating has good metallurgical bonding with the substrate without defects such as pores and cracks [18]. Moreover, the coatings were composed of black equiaxed crystal structure and white reticular intergranular structure. The black equiaxed crystal structure consisted of α-Fe phase and (Fe, Cr) solid solution. The white reticular intergranular structure was composed of (Fe, Cr) solid solution, (Fe, Cr)_7_C_3_ carbide, and (Al, Fe)_4_Cr hard phase [16].

Due to the deficiency of nuclei and the small molten pool, the solidification process showed a strong epitaxial growth tendency from the substrate to the coating. This led to the formation of coarse columnar dendrites perpendicular to the scanning direction, which caused anisotropy, impaired the cladding quality, and increased the probability of flaking [19]. When the columnar dendrite decreases, the anisotropy of the coating is reduced and the bonding performance of the coating is increased [20]. In the sample of coating A (Figure 4(a1)), coarse columnar dendrites grew above the planar dendrites, and the coarse equiaxed dendrites followed.

As seen in Figure 4(b1), the size of columnar dendrites decreased gradually and the proportion of equiaxed dendrites increased in the bottom of coating B. Moreover, all of the columnar dendrites were separated from the bright planar crystal. The Lorentz force (F_L_) generated by EF aggravated the Marangoni convection (MC) at the bottom, which increased the stress at the root of the twin columnar dendrites, causing the fracture and formation of the independent fine columnar dendrites. Figure 5a shows the columnar dendrite fracture mechanism under EF, and Figure 5b shows its schematic diagram.

As shown in Figure 4(c1), the grain size in the bottom region of coating C began to coarsen according to the following formula:(1)$$\upsilon *\rho^{2}=\frac{8\pi^{2}D_{L}\Gamma }{m_{L}C_{0}\left(k_{0}-1\right)},$$
where υ is the growth rate, ρ is the tip radius of the dendrite, $D_{L}$ is the temperature of the liquid, Γ is the Gibbs–Thompson parameter, $m_{L}$ is the slope of the liquidus, $C_{0}$ is the initial composition, and $k_{0}$ is the equilibrium partition coefficient. The product of υ and $\rho^{2}$ was constant during the grain growth [21]. When the EFS was 40 mT, a large amount of Joule heat rapidly accumulated on the tip of the dendrite, the undercooling significantly declined, and the growth rate decreased. With the decreased growth rate, grain fusion occurred, and a large tip radius was observed (Figure 4(c1)). This implied that, when the EFS was 40 mT, Joule heat had a greater effect on coarsening grains than the Lorentz force refinement effect.

In Figure 4(a2–c2), the refinement law of cells and equiaxed dendrites in the middle upper region under the effect of EF was the same as that of the bottom region. The grain refinement here could be divided into two mechanisms:(1)Joule heat was produced upon inducing current in the molten matrix, which increased the temperature and prolonged the nucleation time when the temperature fell below the solidus temperature. Longer time was available to nucleate, and the quantity of nuclei improved [21].(2)The external electromagnetic energy increased the total kinetic energy in the molten pool. The F_L_ generated by the EF could enhance the MC and decrease the temperature gradient in the molten pool. According to the rapid solidification theory [22], the temperature gradient G and the solidification rate R have a decisive effect on the grain size. When the temperature gradient decreased, the G/R ratio also decreased, which promoted the minification of the grain size.

The relationship linking dendrite secondary arm spacing λ_2_, cooling speed V, and temperature gradient G is as follows [23]:(2)$$\lambda_{2}=\alpha {\left(G\times V\right)}^{-\delta },$$
where α and δ are constants related to the material. A smaller temperature gradient resulted in larger secondary dendrite arm spacing. According to the dendrite arm remelting theory [24], the arms of equiaxed dendrites were more easily broken by molten metal and then distributed as nuclei in the molten pool [25].

Figure 6 shows the line scanning images of Fe, Cr, and Al elements at the interface with and without EF. The bonding interface is a region, and its size and element distribution reflect the bonding properties between the substrate and the coating. It can be seen that the Fe element was increased from the coating to the substrate, while Cr had the opposite trend. During the LC, the substrate and powder were heated and melted under laser irradiation to form a molten pool. All elements in the substrate and powder had a certain diffusion ability [26]. With EF assistance, the shape of the bonding interface fluctuated more obviously. After measurement, the bonding interfaces of coatings A, B, and C were 25.19 μm, 19.79 μm, and 26.99 μm, respectively. Furthermore, the transition of the scan line spectrum of Fe and Cr in the bonding interface region was more gentle, as shown in Figure 6b,c. This indicates that the application of 20 mT EF promoted the uniform distribution of elements in the bonding interface region and reduced the size to improve the interface bonding performance [27].

Figure 7 represents the XRD diffraction patterns of coatings A, B, and C. It was found that the phases mainly included (Fe, Cr) solid solution, (Fe, Cr) _7_C_3_ carbide, Fe_3_Al, and (Al, Fe)_4_Cr. Al_2_O_3_ ceramic particles were not detected by XRD, due to the melting of the Al_2_O_3_ ceramic particles in the composite coating, where they combined with Fe and Cr to form Fe_3_Al and (Al, Fe)_4_Cr intermetallic compounds [16,28]. Obviously, the addition of EF only changed the phase content of the coating, but did not change the phase composition.

### 3.2. Microhardness and Volume Loss

Figure 8 shows the average microhardness and volume loss (VL) of coatings A, B, and C. The average microhardness of coatings A, B, and C was 782.2 HV_0.2_, 866.7 HV_0.2_, and 814.7 HV_0.2_ (standard deviations of 10.9, 9.5, and 10.5), respectively, showing the opposite of the tendency toward VL. The average VL of coatings A, B, and C was 57.8 × 10^−3^ mm^3^, 20.9 × 10^−3^ mm^3^, and 45.2 × 10^−3^ mm^3^ (standard deviations of 0.86, 0.67, and 0.81), respectively. This confirms that higher coating microhardness results in better wear resistance and less VL [29]. The microhardness of coating B was 84.5 HV_0.2_ higher than that of coating A, whereas it was just 32.5 HV_0.2_ higher compared to coating C. This indicates that the microhardness of coating B was significantly improved.

The microhardness is closely related to microstructure and phase composition [30]. According to the analysis of microstructure and phase composition, the EF assistance made the microstructure more homogeneous and denser, with more hard phases, which jointly improved the coating microhardness. This implies that EF could play a beneficial role in improving the performance of coatings. With EF assistance, the VL generally decreased. The VL of coatings B and C was lower by factors of 63.8% and 21.8% compared to coating A. These findings again confirm that only the application of a proper EF in LC could significantly improve the wear resistance of coatings.

### 3.3. Friction Coefficient

Figure 9 shows the friction coefficient of coatings A, B, and C. Taking coating B as an example, the friction coefficient was divided into three parts: the running-in period, the fast-rising period, and the stationary period [31].

The duration of the running-in period was the longest for coating B. The friction coefficient was extremely small (about 0.14) before it entered the fast-rising period, which lasted for nearly 300 s. Figure 4(b1,b2) can explain this phenomenon, whereby the finer cells and equiaxed dendrites in the middle upper region generated the highest microhardness, which enabled the coating to resist the intrusion of the Si_3_N_4_ ceramic ball for a longer time before the matrix was abased. Early in the stationary period, the friction coefficient of all coatings was uniform. The difference between the friction coefficients gradually formed after 800 s. In this period, the grain refinement and the high content of hard phases in coating B showed a better anti-friction effect [32]. In contrast to coating A, the wear resistance of coating C was weakened. According to Figure 4(c1,c2) and the XRD analysis, this could be ascribed to the coarsening of the microstructure and the decrease in hard phases in the middle upper region of the coating.

### 3.4. Wear Scars

Figure 10 presents partial SEM images of the worn surface and wear debris of coatings A, B, and C. As can be seen in Figure 10(a1), there were a large number of spalling zones, in addition to severe plastic deformation and local micro-flaking. Meanwhile, the flat debris peeled off from the coating can be seen in Figure 10(a2), exhibiting the severe adhesive wear seen on coating A. There were a large number of ploughing grooves on the worn surface of coating A, indicating abrasive wear [33].

As shown in Figure 10(b1), coating B was smoother with a slight ploughing groove, no cracks, and no accumulated adhesive layer. This suggests that the main wear mechanism of coating B was abrasive wear, accompanied by slight adhesive wear, which could significantly improve the wear resistance of the coating [33]. Furthermore, the size of debris in Figure 10(b2) was the smallest. This also shows that the wear performance of coating B was best [34]. This can be credited to the increase in hard phase in the coating, which significantly enhanced the microhardness and prevented plowing, while the grain refinement enhanced the toughness of the coating, prevented the initiation and propagation of cracks, and inhibited spalling [35,36].

As shown in Figure 10(c1), there were a large number of ploughing grooves, a few peeling areas, and slight plastic deformation on the worn surface of coating C. Compared with coating A, the main wear mechanism of coating C was abrasive wear, and the wear resistance was improved [37]. However, compared with coating B, the ploughing grooves in coating C were more obvious, the size of wear debris was significantly increased, and flat debris appeared, as seen in Figure 10(c1,c2). This indicates that the wear resistance of coating C was lower than that of coating B [33,34]. This can be attributed to the grain coarsening in the upper middle part of coating C, which reduced the hardness and plasticity of the coating. The results show that the use of EF could improve the wear resistance of the Fe901/Al_2_O_3_ MMC coatings, but the effect was weakened when the EFS was 40 mT. Therefore, in the LC process, a coating with higher wear resistance can be prepared on the surface of gears and spindles by using appropriate EFS to improve their service life.

## 4. Conclusions

In this paper, Fe901/Al_2_O_3_ MMC coatings were fabricated on a 45 steel substrate by LC with an EF. The following conclusions can be drawn:(1)The EF generated Lorentz force and a Joule heating effect, which increased the nucleation rate of grains, strengthened the MC flow, broke the coarse columnar dendrites, and refined the coating structure. However, when EFS was 40 mT, the Joule heat effect had a greater impact than Lorentz force, resulting in a reduction in coating quality. The optimal EFS was 20 mT, resulting in an average microhardness of 866.7 HV_0.2_ and a volume loss pf 20.9 × 10^−3^ mm^3^.(2)The application of EF did not change the phase composition of the coating; the coatings with and without EF contained (Fe, Cr), (Fe, Cr)_7_C_3_, Fe_3_Al, and (Al, Fe)_4_Cr phases. The increase in content of (Fe, Cr) and (Fe, Cr)_7_C_3_ hard phase enhanced the wear resistance of the Fe901/Al_2_O_3_ MMC coatings.(3)The wear mechanism of the Fe901/Al_2_O_3_ MMC coating without EF was severe adhesive wear accompanied by slight abrasive wear. The application of EF switched the main wear mechanism of Fe901/Al_2_O_3_ MMC coatings from adhesive wear to abrasive wear, which improved the wear resistance of the coating, because the wear surface had a small ploughing groove, the plastic deformation was reduced, and there was no large peeling debris.

## Figures and Tables

**Figure 1 materials-15-01531-f001:**
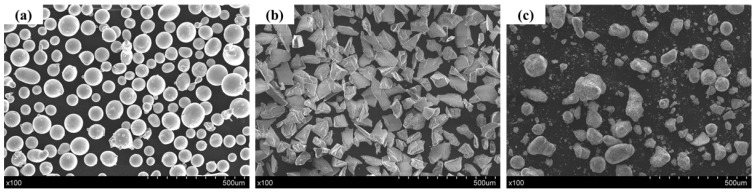
SEM images of powder: (**a**) Fe901 powder; (**b**) Al_2_O_3_ powder; (**c**) Fe901/Al_2_O_3_ mixed powder.

**Figure 2 materials-15-01531-f002:**
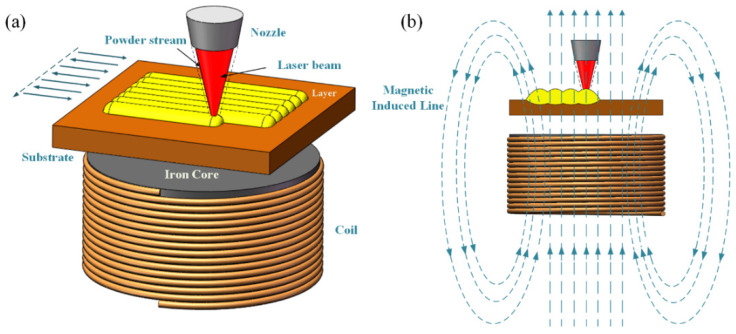
(**a**) Schematic diagram of LC with EF, (**b**) schematic diagram electromagnetic field distribution.

**Figure 3 materials-15-01531-f003:**
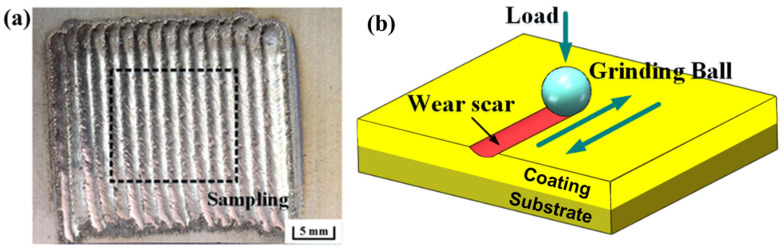
(**a**) Surface morphology of the coating; (**b**) schematic diagram of wear test.

**Figure 4 materials-15-01531-f004:**
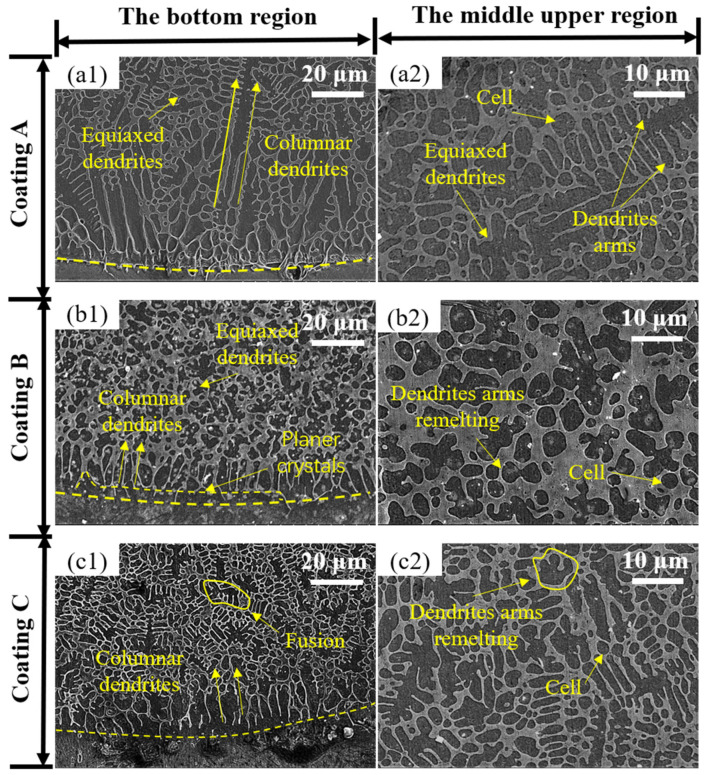
The microstructure of coatings at the bottom (**a1**–**c1**) of coatings A (0 mT), B (20 mT), C (40 mT) and in the middle upper region (**a2**–**c2**) of coatings A (0 mT), B (20 mT), C (40 mT).

**Figure 5 materials-15-01531-f005:**
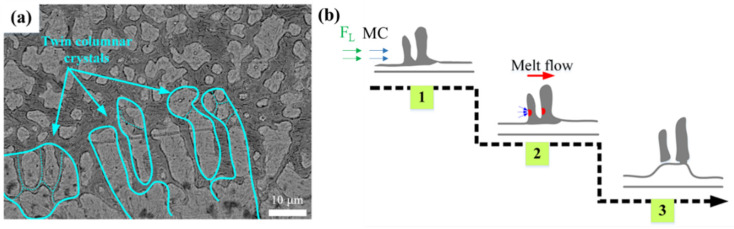
(**a**) Columnar dendrite fracture mechanism of coating B (20 mT) and its (**b**) schematic diagram.

**Figure 6 materials-15-01531-f006:**
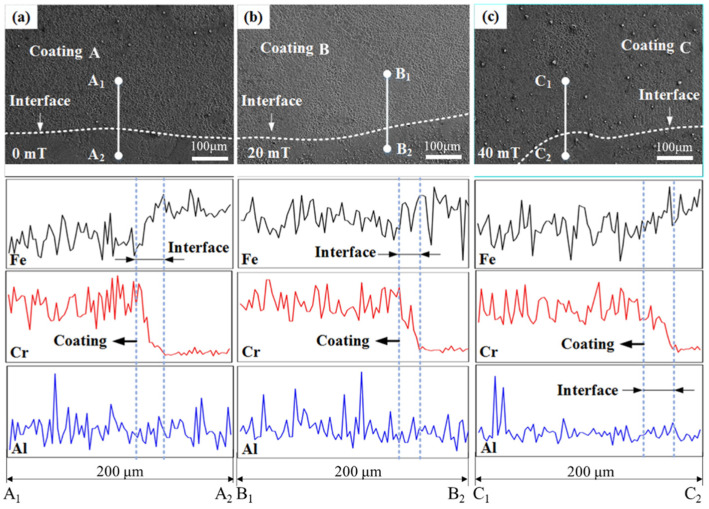
SEM images of metallurgical bonding and the line scanning images of coatings (**a**) A (0 mT), (**b**) B (20 mT), and (**c**) C (40 mT).

**Figure 7 materials-15-01531-f007:**
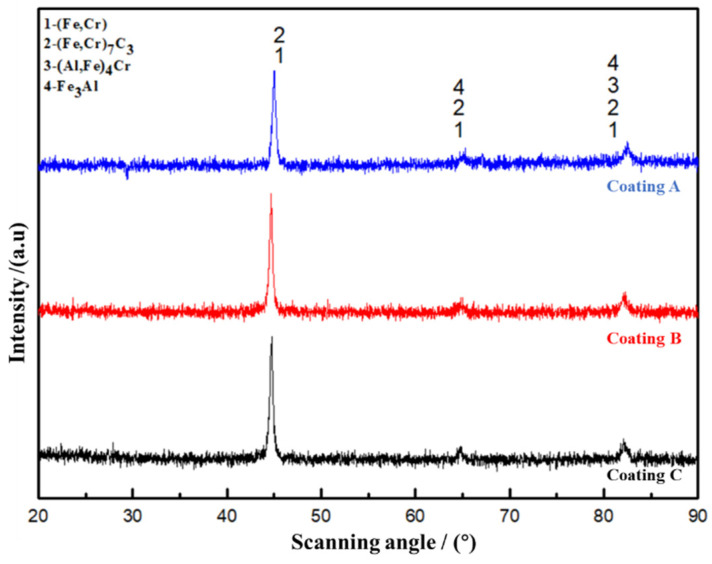
XRD diffraction pattern of coatings A (0 mT), B (20 mT), and C (40 mT).

**Figure 8 materials-15-01531-f008:**
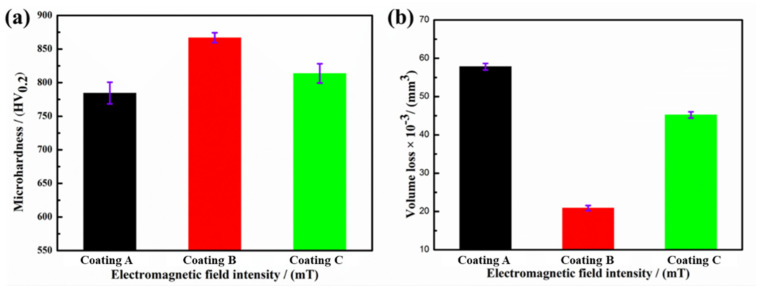
Average microhardness (**a**) and volume loss (**b**) of coatings A (0 mT), B (20 mT), and C (40 mT).

**Figure 9 materials-15-01531-f009:**
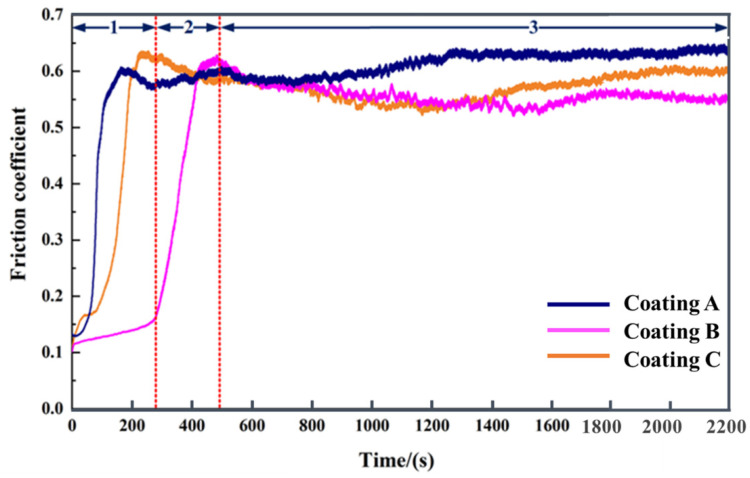
The friction coefficient of coatings A (0 mT), B (20 mT), and C (40 mT) (1: the running-in period, 2: the fast-rising period, 3: the stationary period).

**Figure 10 materials-15-01531-f010:**
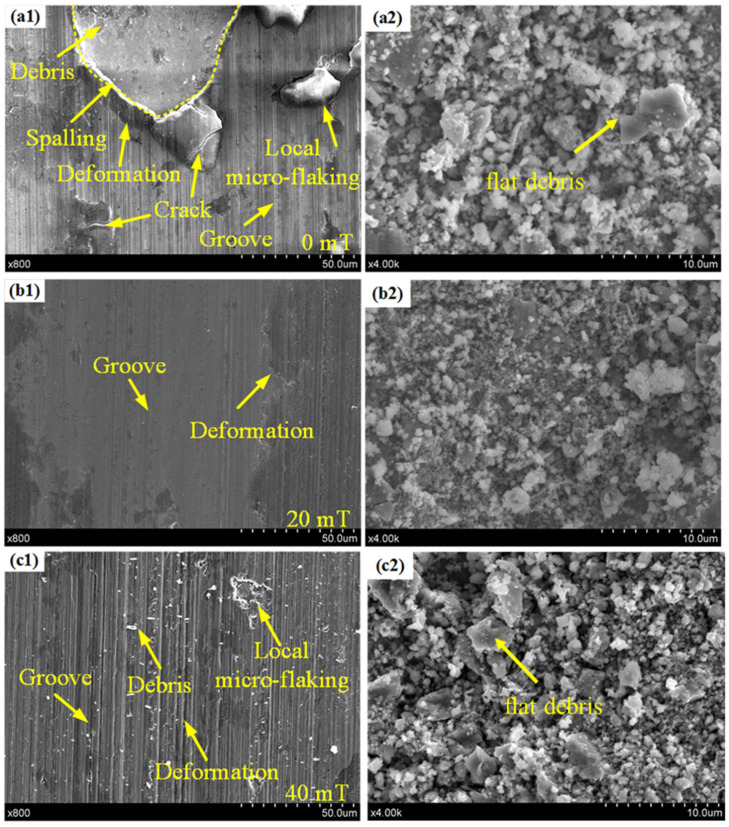
Comparison of local morphology of wear scars (**a1**–**c1**) of coating A (0 mT), B (20 mT), and C (40 mT); and wear debris (**a2**–**c2**) of coatings A (0 mT), B (20 mT), and C (40 mT).

**Table 1 materials-15-01531-t001:** Chemical composition of 45 steel substrate.

Element	C	Mn	B	Si	Fe
Composition (wt.%)	0.45	0.59	1.6	0.25	Bal.

**Table 2 materials-15-01531-t002:** Chemical composition of Fe901 iron-based alloy powder.

Element	C	Cr	B	Si	Mo	Fe
Composition (wt.%)	0.15	13.6	1.6	1.2	0.8	Bal.

**Table 3 materials-15-01531-t003:** Commercial sources for the raw materials.

45 Steel Substrate	Shanghai Moju Special Steel Co., Ltd., China
Fe901 powder	Beikuang Xincai Technology Co., Ltd., China
Al_2_O_3_ powder	Beikuang Xincai Technology Co., Ltd., China

**Table 4 materials-15-01531-t004:** Processing parameters for LC.

	Laser Power (W)	Scanning Speed (mm/min)	Powder Feeder Rate (g/min)	Shield Gas Flow (L/min)	Spot Diameter (mm)	Overlap Rate (%)	EFS (mT)
Coating A	1600	400	9.5	15	3	20	0
Coating B							20
Coating C							40

## Data Availability

Not applicable.

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
