# Peer review of "Effect of Electromagnetic Field on Wear Resistance of Fe901/Al2O3 Metal Matrix Composite Coating Prepared by Laser Cladding"

_materials, 2022, doi:10.3390/ma15041531_

Round 1
Reviewer 1 Report
In this interesting study, electromagnetic field -assisted laser cladding technology was used to prepare Fe901/Al2O3 MMC coatings with higher wear resistance. The authors explored the influence of electromagnetic field on microstructural evolution, phase composition, and wear resistance of obtained metal matrix composite coatings.
The manuscript could be accepted after corrections of minor methodological errors and text editing.
Concretely:
-Line 38 introduction MMC term should be defined before using the acronym
-Line 41 TIC should be TiC?
-Line 99 Twenty points were randomly selected from each LC sample, and the average 99 value was adopted. A standard deviation should be also calculated.
-Figure 3 better delimitate the interface between coating and substrate
-Line 193 The Al2O3 ceramic particles were not detected by XRD, indicated that the Al2O3 completely dissolved into the Fe matrix, forming the Fe3Al and (Al, Fe)4Cr intermetallic. A reference to support this assumption is required
-Line 202 VL should be defined as volume loss
-Some references for Friction coefficient discussion part are needed.
-The EF generated Lorentz force and Joule heating effect which increased the nucleation rate of grains, strengthened the MC flow, broken the coarse columnar dendrites, and refined the coating structure. However, when EFS exceeded, the Joule heat effect had a greater impact than Lorentz force, resulting in the reduction of coating quality. Could you define the optimized parameters in conclusion part?
- The readers would appreciate more details about possible industrial application of the EF switched the main wear mechanism of Fe901/Al2O3 MMC coating from adhesive wear to abrasive wear.

Author Response
Dear Reviewer
On behalf of my co-authors, we thank you very much for giving us an opportunity to revise our manuscript, we appreciate editor and reviewers very much for their positive and constructive comments and suggestions on our manuscript entitled “Effect of electromagnetic field on wear resistance of Fe901/Al2O3 metal matrix composite coating prepared by laser cladding”. (Manuscript ID: materials-1588007).
We have studied reviewer’s comments carefully and have made revision which marked in red in the paper. We have tried our best to revise our manuscript according to the comments. Attached please find the revised version, which we would like to submit for your kind consideration.
We would like to express our great appreciation to you and reviewers for comments on our paper. Looking forward to hearing from you.
Thank you and best regards.
Sincerely yours,
Yaobang Chen
Responds to the reviewer’s comments:
Reviewer :
In this interesting study, electromagnetic field -assisted laser cladding technology was used to prepare Fe901/Al2O3 MMC coatings with higher wear resistance. The authors explored the influence of electromagnetic field on microstructural evolution, phase composition, and wear resistance of obtained metal matrix composite coatings.
The manuscript could be accepted after corrections of minor methodological errors and text editing.
1. Line 38 introduction MMC term should be defined before using the acronym
Response 1: line 39 Thank you for your suggestion. We have defined MMC term before using the acronym. 2. Line 41 TIC should be TiC?
Response 2: line 43
Thank you very much for your reminder. We have modified the spell of TiC in the revised manuscript.
- Line 99 Twenty points were randomly selected from each LC sample, and the average value was adopted. A standard deviation should be also calculated.
Response 3: line 115 and lines 226 to 227 Thanks for your suggestion. We have added the calculation of the standard deviation in text in line 115 according to the formula:. The standard deviation of microhardness values of coatings A, B and C are 10.9, 9.5 and 10.5, respectively (in lines 226 to 227).
4. Figure 3 better delimitate the interface between coating and substrate
Response 4: Figure 3
We very much agree with your suggestion. We have modified the color between the coating and the substrate to delimitate them better in Figure 3.
- Line 193 The Al2O3 ceramic particles were not detected by XRD, indicated that the Al2O3 completely dissolved into the Fe matrix, forming the Fe3Al and (Al, Fe)4Cr intermetallic. A reference to support this assumption is required
Response 5: line 218
Thank you very much for your good advice. After your reminder, we have cited references (Surface and Coatings Technology 238 (2014) 9-14; Acta Optica Sinica 39 (2019) 0514001) in the revised manuscript. Due to the melting of the Al2O3 ceramic particles in the composite coating, they will combine with Fe and Cr to form Fe3Al and (Al, Fe)4Cr intermetallic compounds. And we also did not detect Al2O3 via XRD, which proves the above statement.
6. Line 202 VL should be defined as volume loss
Response 6: line 225
Thanks for your suggestion. We have defined VL term in line 225 as volume loss before using acronym.
7. Some references for Friction coefficient discussion part are needed.
Response 7: lines 247 and 260
Thank you for your valuable suggestion. After your reminder, we have cited references (Wear 476 (2021) 203723; Journal of Materials Processing Tech(2020) 116671) for Friction coefficient discussion part in the revised manuscript.
- The EF generated Lorentz force and Joule heating effect which increased the nucleation rate of grains, strengthened the MC flow, broken the coarse columnar dendrites, and refined the coating structure. However, when EFS exceeded, the Joule heat effect had a greater impact than Lorentz force, resulting in the reduction of coating quality. Could you define the optimized parameters in conclusion part?
Response8: lines 304 to 305
Thank you very much for your valuable suggestions. We have added the optimized parameters in conclusion part.
- The readers would appreciate more details about possible industrial application of the EF switched the main wear mechanism of Fe901/Al2O3 MMC coating from adhesive wear to abrasive wear.
Response 9: lines 291 to 293
Thank you for your advice, which is very important to us. We have added the possible industrial application of the EF that the assistance of EF improves the wear resistance of the coating, which can be used in the surface strengthening of gears and spindles.
Special thanks to you for your good comments.

Reviewer 2 Report
This work explores how an external electromagnetic field applied during laser cladding can improve the properties of the coating to increase its resistance to wear and potentially increase the service life of the product. The strengths of this article include a clear experimental method that could be replicated without issue, and the liberal use of schematics for the sake of illustrating concepts and procedures. However, the data analysis and interpretation should be more quantitative- such details as how many times a measurement was performed for statistical purposes was missing.
Specific comments to improve the manuscript follow:
- The second-last paragraph of the introduction (lines 37 to 47) provides a review of previous EF-assisted laser processing applications but was difficult to follow in some places due to the lack of comparison with controls having no EF component. For example, the sentence on lines 41-43 informs that EF “greatly improved its surface hardness and wear resistance”, but what is the control? By how much did these properties improve relative to the procedure without EF? Moreover, three of the four references for EF processing being “widely used” are from the authors’ group. Additional references from other researchers should be added here to justify the broad use of EF techniques.
- It’s odd that Table 1 would exclude the substrate composition information. If that data can just be written in the text, why not just also write the composition for the alloy? Or why not put both materials in the table?
- Commercial sources for the raw materials should be added
- Figure 1 is interesting but requires more elaboration to justify its presence. How is the morphology important? If everything is milled and made to be somewhat jagged, how much does the original jaggedness of the ceramics contribute as “reinforcement”? If the authors wish to keep Figure 1, SEM of the milled mixture should be added.
- It is not explained why the parameters in Table 2 are optimal. Were other parameters attempted? Are these parameters from a previous experiment, in which case the reference should be cited? Also, why were 20 and 40 mT electromagnetic field strengths chosen?
- When describing the wear test, it was stated that multiple cladding layers were necessary, because one wasn’t enough. On line 102, it is stated that the sample thickness is 6 mm, so were two passes used to make the sample thicker? Is that because the ceramic ball would scrape clear through to the substrate if only one layer was present? Does the cladding adhere differently to itself than to the substrate, and does this interface between the substrate and the cladding versus the cladding and itself have any impact on the test? It would be nice to add a third panel to Figure 3 showing the cladding after completion of the wear test to see the difference with the image in Figure 3a.
- The text on line 117 is not a complete sentence, and the contention that specific regions of the cladding react to the wear test should be justified with a reference.
- In Figure 4, it’s difficult to see the magnification and scale-bar of each image.
- Throughout page 5, when discussing Figures 4 and 5, a lot of structural details are mentioned and described, but it is rarely clarified what the presence and distribution of these structures means. For example, line 130 says that in sample B, the columnar dendrites gradually become smaller as compared to sample A, but it is not clear from looking at Figure 4 that this is the case. It would help to highlight a specific region of panels A and B to show this difference. Moreover, some explanation of why reduced columnar dendrite size is good would be helpful to justify why the EF condition producing B is optimal.
- The discussion about Joule heating beginning on line 140 is difficult to follow. First, Formula 3 is mentioned for the first time in line 147 but doesn’t actually appear until 171; it’s actually mentioned before Formula 2 is mentioned or shown. Second, no direct relationships between the three equations and experimental data are drawn. There are several relationships mentioned (like the solidus temperature and nucleation time, the temperature gradient and grain size), but it is difficult, especially for non-experts, to fit everything together when this discussion is disconnected from the experimental data. Where do the actual applied EM field values fit into this? Can it be shown mathematically why Joule heat became the dominant effect when the field was increased over a certain value?
- The line-scanning elemental composition results in Figure 6 are quite noisy: were these analyses only performed once per sample? The assertion about the slope-decrease (in line 185) seems more like a qualitative declaration than a quantitative one with the way the data is presented. How were the reported interface lengths determined? Is the interface length the distance between the pairs of vertical dotted lines on the element line scans? If so, the length in µm should be provided on the x axis for all panels.
- Figure 7 should be discussed more thoroughly. Neither the text nor the caption mention that there are two parts (7a and 7b) to this figure. Where would the Al2O3 peak be if it had shown up? Why is Figure 7b present showing a magnification of the 45 degree peak?
- In Figure 8, are the error bars standard deviations? If so, it would help to provide the standard deviation values in the text because by eye, the error-bars in Figure 8a don’t really imply as much of a difference between the samples as the text is asserting.
- Is the labeling of the three time periods is in Figure 9 meant to apply to sample B (20 mT) only? If so, that should be specified in the text. Moreover, it would help to label the three samples A, B, C as before for clarity. Finally, the caption of Figure 9 should indicate what regions 1, 2, 3 mean (fast-rise, stationary, etc.)
- In the final paragraph of this section about friction coefficients (starting on line 226), it is not clear that in the stationary period B outperformed C, although both clearly outperformed A. Was this analysis only performed once on each sample? Perhaps multiple measurements would make differences between B and C clearer if they exist.
- Where do the error bars in Figure 10d come from? According to that panel, samples A and C look to have performed the same, in contrast to the text that says C performed worse (line 242)
- Do the results of the wear test impact the interpretation of the microhardness and volume-loss tests? Because it looks like C was the softest material and potentially lost the most volume according to Figure 10, but that’s not reflected in Figure 8. It would be helpful to elucidate the interpretation of how these tests interact and inform one another for non-experts.
- The discussion of Figure 11 is confusing. First, the difference between adhesion and abrasive wear should be clearly explained. The explanation for adhesion wear in sample A is elaborated upon with descriptions of what is happening to the sample physically and how those interactions might relate to the sample’s microstructure, although nothing is said to justify to the readers why this would entail one kind of wear and not the other. The explanation for the behavior of sample C is simply that “the electromagnetic field has to be just so for the sample to be strengthened. Therefore, this sample was damaged mainly by abrasive wear”. That assertion is neither helpful nor sufficient. The data in Figure 11 present an opportunity to really contrast the microstructures and it should definitely be explained why one kind of wear dominates over another.
- You draw the following conclusions:
- The application of an electromagnetic field increased grain nucleation rate, strengthened Marangoni convection flow, and broke the coarse columnar dendrites to yield refined microstructure. “Too much” electromagnetic field strength favored Joule heating over Lorentz force, and produced a courser grain.
- The application of an electromagnetic field increased the microhardness of the cladding; with the sample prepared under a 20 mT field performing the best.
- Without the presence of an electromagnetic field, the predominant wear mechanism was adhesive wear, but in the presence of an EM field, the predominant wear mechanism was abrasive wear.
- Conclusion 1 would be more definitive if the “microhardness and volume loss” section was more concrete, clear, and concise.
- For conclusion 2, the evidence is not presented in a reassuring way. It is pertinent to specify how many tests were performed across all investigations, and what the standard errors were. The error-bars in this section should be acknowledged and discussed, and the values should be put in perspective. Finally, the data from this section should be discussed in connection with the results from the sections on the “friction coefficient” and “wear scars”, as they contribute directly to the volume loss data and surely elucidate the microhardness data in some way.
- For conclusion 3, It should be specified how it was determined that wear was either adhesive or abrasive, and the significance should be explained.
Author Response
Dear Reviewer
On behalf of my co-authors, we thank you very much for giving us an opportunity to revise our manuscript, we appreciate editor and reviewers very much for their positive and constructive comments and suggestions on our manuscript entitled “Effect of electromagnetic field on wear resistance of Fe901/Al2O3 metal matrix composite coating prepared by laser cladding”. (Manuscript ID: materials-1588007).
We have studied reviewer’s comments carefully and have made revision which marked in red in the paper. We have tried our best to revise our manuscript according to the comments. Attached please find the revised version, which we would like to submit for your kind consideration.
We would like to express our great appreciation to you and reviewers for comments on our paper. Looking forward to hearing from you.
Thank you and best regards.
Sincerely yours,
Yaobang Chen
Responds to the reviewer’s comments:
Reviewer :
1.The second-last paragraph of the introduction (lines 37 to 47) provides a review of previous EF-assisted laser processing applications but was difficult to follow in some places due to the lack of comparison with controls having no EF component. For example, the sentence on lines 41-43 informs that EF “greatly improved its surface hardness and wear resistance”, but what is the control? By how much did these properties improve relative to the procedure without EF? Moreover, three of the four references for EF processing being “widely used” are from the authors’ group. Additional references from other researchers should be added here to justify the broad use of EF techniques.
Response 1: lines 38 to 49 and line 58
Thank you very much for your valuable comments. We checked the last second paragraph of the introduction (lines 38 to 49). In fact, here is a review of MMC coating, and the use of EF has not been discussed. According to your suggestion, we have added properties improve relative to the procedure without EF in line 58.According to your suggestion, we only keep one reference of our group, and quote other references(Optics and Laser Technology 99 (2018) 342-350; Applied Surface Science 321 (2014) 252–260).
2. It’s odd that Table 1 would exclude the substrate composition information. If that data can just be written in the text, why not just also write the composition for the alloy? Or why not put both materials in the table?
Response 2: Table 1
Thank you very much for your advice. We have added the chemical composition of 45 steel in a new Table 1.
3. Commercial sources for the raw materials should be added
Response 3: Table 3
Thank you very much for your valuable suggestions. We added Table 3 to explain the commercial sources of substrate (Shanghai moju Special Steel Co., Ltd, China) and powder (Beikuang Xincai Technology Co., Ltd, China).
4. Figure 1 is interesting but requires more elaboration to justify its presence. How is the morphology important? If everything is milled and made to be somewhat jagged, how much does the original jaggedness of the ceramics contribute as “reinforcement”? If the authors wish to keep Figure 1, SEM of the milled mixture should be added.
Response 4: Figure 1
I sincerely thank you for your valuable advice. Please allow us to explain: 1. Figure 1 is a visual display of Fe901, Al2O3 and Fe901/Al2O3 (grinding mixture of Fe901 and Al2O3) materials; 2. The morphology of materials does have a crucial impact on the wear resistance, but at present, this paper mainly discusses the effect of EF on the wear resistance; 3. Figure 1 (c) shows the grinding mixture of Fe901 and Al2O3.
5. It is not explained why the parameters in Table 2 are optimal. Were other parameters attempted? Are these parameters from a previous experiment, in which case the reference should be cited? Also, why were 20 and 40 mT electromagnetic field strengths chosen?
Response 5: Table 4 and line 104
Thank you for your kind suggestion. The optimal parameters in Table 4 (Original Table 2) are obtained from our previous work, and the reference (Acta Optica Sinica 39 (2019) 0514001) has been cited in the revised manuscript. We have tried electromagnetic field strengths of 0, 10, 20, 30, and 40 mT respectively. The results are shown in the following Figure S1, 0, 20 and 40 mT are representative data.
Figure. S1 The friction coefficient of MMC coatings under EF.
6. When describing the wear test, it was stated that multiple cladding layers were necessary, because one wasn’t enough. On line 102, it is stated that the sample thickness is 6 mm, so were two passes used to make the sample thicker? Is that because the ceramic ball would scrape clear through to the substrate if only one layer was present? Does the cladding adhere differently to itself than to the substrate, and does this interface between the substrate and the cladding versus the cladding and itself have any impact on the test? It would be nice to add a third panel to Figure 3 showing the cladding after completion of the wear test to see the difference with the image in Figure 3a.
Response 6: lines 119 to 120
Thank you for your valuable comments. The thickness of our cladding layer is 1mm, which does not need secondary cladding and wrapping, so that the coating can not be worn when the ceramic ball is rubbed. The sample thickness expressed in this paper is 6 mm, which is the total thickness of substrate and cladding layer. The above problems have been revised in the revised manuscript, respectively in lines 119 to 120.
As you described, the adhesion of the cladding layer to itself is indeed different from that of the substrate. Since the substrate and the cladding layer are metallurgically bonded, the adhesion between the substrate and the cladding layer is very strong and will not affect the friction process.
Thank you very much for your suggestion. We have indeed photographed the 3D outline of the cladding after the wear test. We are very ashamed that we have omitted to consider the shooting of the real object for comparison. At present, the real object map cannot be provided for various reasons, and will be added later if necessary.
- The text on line 117 is not a complete sentence, and the contention that specific regions of the cladding react to the wear test should be justified with a reference.
Response 7: lines 134 to 135
Thank you very much for your valuable comments. We have corrected this error in line 134 and cited a reference (Wear 260 (2006) 838–846).
- In Figure 4, it’s difficult to see the magnification and scale-bar of each image.
Response 8: Figure 4
Thank you very much for your suggestion. We have added scale-bar in Figure 4.
9. Throughout page 5, when discussing Figures 4 and 5, a lot of structural details are mentioned and described, but it is rarely clarified what the presence and distribution of these structures means. For example, line 130 says that in sample B, the columnar dendrites gradually become smaller as compared to sample A, but it is not clear from looking at Figure 4 that this is the case. It would help to highlight a specific region of panels A and B to show this difference. Moreover, some explanation of why reduced columnar dendrite size is good would be helpful to justify why the EF condition producing B is optimal.
Response 9: lines 139-143 and lines 148 to 150
Thank you very much for your suggestion. We have added a description of the microstructure composition in lines 139-143. The black equiaxed crystal structure consists of α-Fe phase and (Fe, Cr) solid solution. The white reticular intergranular structure is composed of (Fe, Cr) solid solution, (Fe, Cr)7C3 carbide and (Al, Fe)4Cr hard phase. As you suggested, we have described in lines 148 to 150 that the reduction of columnar crystals can improve the coating quality.
10. The discussion about Joule heating beginning on line 140 is difficult to follow. First, Formula 3 is mentioned for the first time in line 147 but doesn’t actually appear until 171; it’s actually mentioned before Formula 2 is mentioned or shown. Second, no direct relationships between the three equations and experimental data are drawn. There are several relationships mentioned (like the solidus temperature and nucleation time, the temperature gradient and grain size), but it is difficult, especially for non-experts, to fit everything together when this discussion is disconnected from the experimental data. Where do the actual applied EM field values fit into this? Can it be shown mathematically why Joule heat became the dominant effect when the field was increased over a certain value?
Response 10:
Thank you very much for your valuable suggestions. After careful consideration, we decided to discuss this part again. The electromagnetic field intensity used in the experiment is consistent with the reality. For example, the electromagnetic field intensity used in the reference (Surface & Coatings Technology 359 (2019) 125–131) is 47 mT.
- The line-scanning elemental composition results in Figure 6 are quite noisy: were these analyses only performed once per sample? The assertion about the slope-decrease (in line 185) seems more like a qualitative declaration than a quantitative one with the way the data is presented. How were the reported interface lengths determined? Is the interface length the distance between the pairs of vertical dotted lines on the element line scans? If so, the length in µm should be provided on the x axis for all panels.
Response 11: Figure 6
Thank you very much for your valuable advice, which is very useful to us. Firstly, the line scanning images in the article are the results of many experiments, and the relevant references are compared and consulted. (Advanced Composites and Hybrid Materials (2021) 4:205–211; Journal of Alloys and Compounds 505 (2010) 645–653). We think what you said about the lack of quantitative description of slope-decrease is very correct. After consideration, we decided to delete this part of the imprecise statement. The interface length in this paper is obtained by measurement according to the mutation of Cr, and Fe elements, with reference to the literature (Journal of Alloys and Compounds 505 (2010) 645–653). The dotted line in the Figure 6 is marked as the interface length. According to your suggestion, a line scanning length is added to the Figure 6.
- Figure 7 should be discussed more thoroughly. Neither the text nor the caption mention that there are two parts (7a and 7b) to this figure. Where would the Al2O3 peak be if it had shown up? Why is Figure 7b present showing a magnification of the 45 degree peak?
Response 12: Figure 7
Thank you very much for your suggestions. We have added a discussion on lines 216 to 218. Relevant references (Acta Optica Sinica 39 (2019) 0514001) shows that Al2O3 will appear between 30° and 40°(Figure S2). Figure 7 (b) is a partial enlarged view of XRD. After your prompt, we realize that this is not necessary, so Figure 7 (b) is removed.
Figure S2
- In Figure 8, are the error bars standard deviations? If so, it would help to provide the standard deviation values in the text because by eye, the error-bars in Figure 8a don’t really imply as much of a difference between the samples as the text is asserting.
Response 13: lines226 to 230
Thank you very much for your advice. After your reminder, we found that there was an error in the deviation calculation by comparing the original data. We have provided the values in Figure 8 and its standard deviation in lines 226 to 230.
- Is the labeling of the three time periods is in Figure 9 meant to apply to sample B (20 mT) only? If so, that should be specified in the text. Moreover, it would help to label the three samples A, B, C as before for clarity. Finally, the caption of Figure 9 should indicate what regions 1, 2, 3 mean (fast-rise, stationary, etc.)
Response 14: Figure 9
Thank you very much for your valuable advice. The labels for the three time periods in Figure 9 are applicable to sample B because the friction coefficient of coating B is obvious, which we use as an example, and we have specified in the text in line 245. According to your suggestion, the label of coatings A, B and C in Figure 9 have been added. In addition, we have added the mean of the description of regions 1, 2 and 3 in the title of Figure 9. 15. In the final paragraph of this section about friction coefficients (starting on line 226), it is not clear that in the stationary period B outperformed C, although both clearly outperformed A. Was this analysis only performed once on each sample? Perhaps multiple measurements would make differences between B and C clearer if they exist.
Response 15: Figure 9
Thank you for your valuable advice. Since friction coefficients data is maintained every 0.001 s, we only took the data of the 1800 s in original manuscript. Now we have adopted your suggestion to supplement the 2200 s image.
- Where do the error bars in Figure 10d come from? According to that panel, samples A and C look to have performed the same, in contrast to the text that says C performed worse (line 242) Do the results of the wear test impact the interpretation of the microhardness and volume-loss tests? Because it looks like C was the softest material and potentially lost the most volume according to Figure 10, but that’s not reflected in Figure 8. It would be helpful to elucidate the interpretation of how these tests interact and inform one another for non-experts.
Response 16:
Thank you very much for your advice. We found that the obviously wrong data was not removed during data processing, resulting in errors in the calculation results (including standard deviation). Now it has been recalculated and changed in Figure 10. Since the content of Figure 10 has been used in volume loss calculation, after thinking, we decided to delete it. Please allow us to delete this part.
- The discussion of Figure 11 is confusing. First, the difference between adhesion and abrasive wear should be clearly explained. The explanation for adhesion wear in sample A is elaborated upon with descriptions of what is happening to the sample physically and how those interactions might relate to the sample’s microstructure, although nothing is said to justify to the readers why this would entail one kind of wear and not the other. The explanation for the behavior of sample C is simply that “the electromagnetic field has to be just so for the sample to be strengthened. Therefore, this sample was damaged mainly by abrasive wear”. That assertion is neither helpful nor sufficient. The data in Figure 11 present an opportunity to really contrast the microstructures and it should definitely be explained why one kind of wear dominates over another.
Response 17: lines 267 to 293
Thank you very much for your important advice. We have revisited Figure 10 (original figure 11) in combination with your suggestion. We explained the of adhesive wear in coating A and explained the possible relationship between these interactions and the microstructure of the coatings. We also reanalyzed sample C and explained that one wear is more dominant than another.
- Conclusion 1 would be more definitive if the “microhardness and volume loss” section was more concrete, clear, and concise.
Response 18: lines 304 to 305
Thank you very much for your valuable advice. We have added the microhardness and wear volume at 20 mT in conclusion 1.
- For conclusion 2, the evidence is not presented in a reassuring way. It is pertinent to specify how many tests were performed across all investigations, and what the standard errors were. The error-bars in this section should be acknowledged and discussed, and the values should be put in perspective. Finally, the data from this section should be discussed in connection with the results from the sections on the “friction coefficient” and “wear scars”, as they contribute directly to the volume loss data and surely elucidate the microhardness data in some way.
Response 19:
Thank you very much for your advice. We found that the obviously wrong data were not removed during data processing, resulting in errors in the calculation results (including standard deviation). Now it has been recalculated and changed.
- For conclusion 3, It should be specified how it was determined that wear was either adhesive or abrasive, and the significance should be explained.
Response 20: lines 311 to 315
Thank you very much for your advice. We have explained the reason for the transition from adhesive wear to abrasive wear in conclusion 3. And added the significance of this transformation.
Special thanks to you for your good comments.

Reviewer 3 Report
The review concerns the manuscript titled "Effect of electromagnetic field on wear resistance of Fe901/Al2O3 metal matrix composite coating prepared by laser cladding"
My comments to this manuscript are below:
1. The Authors should prepare list of abbreviations, which will be use these throughout the manuscript.
2. Section "Introduction" should be supplemented with current items of literature concern the subject of the manuscript, the authors from different countries and continents. Thus, the literature should be completed.
3. Section "Materials and methods"
I suggest adding the characteristics of the material - coatings A, B and C (for example in table form).
4. Section "Results and discussion"
Figure 4. the images in figure c1 and c2 should be for coating C, but the description on the left is "Sample of B with 40 mT EF". Moreover, the descriptions in this figure are unreadable.
Figure 5. The descriptions in this figure are unreadable too.
Figure 8 and Figure 9. The name of coatings are missed. I propose to introduce the designations of the tested samples in section "Materials and methods" and use these designations throughout the manuscript.
Figure 10. The title of this figure is incorrect. The images present the surface morphology with wear traces (!). Please change it.
Author Response
Dear Reviewer
On behalf of my co-authors, we thank you very much for giving us an opportunity to revise our manuscript, we appreciate editor and reviewers very much for their positive and constructive comments and suggestions on our manuscript entitled “Effect of electromagnetic field on wear resistance of Fe901/Al2O3 metal matrix composite coating prepared by laser cladding”. (Manuscript ID: materials-1588007).
We have studied reviewer’s comments carefully and have made revision which marked in red in the paper. We have tried our best to revise our manuscript according to the comments. Attached please find the revised version, which we would like to submit for your kind consideration.
We would like to express our great appreciation to you and reviewers for comments on our paper. Looking forward to hearing from you.
Thank you and best regards.
Sincerely yours,
Yaobang Chen
Responds to the reviewer’s comments:
Reviewer :
- The Authors should prepare list of abbreviations, which will be use these throughout the manuscript.
Response 1: line 314
Thank you very much for your valuable advice. We have added a list of abbreviations at the end of the article.
- Section "Introduction" should be supplemented with current items of literature concern the subject of the manuscript, the authors from different countries and continents. Thus, the literature should be completed.
Response 2:
Thanks for your suggestion. We refer to your valuable comments and have quoted relevant literature from other countries in the introduction.
- Section "Materials and methods" I suggest adding the characteristics of the material - coatings A, B and C (for example in table form).
Response 3: Table 4
Thank you for your suggestion, we agree with it very much. We have added the characteristics of the material - coatings A, B and C in Table 4.
- Section "Results and discussion" Figure 4. the images in figure c1 and c2 should be for coating C, but the description on the left is "Sample of B with 40 mT EF".
Response 4: Figure 4
Thank you very much for your valuable advice. We have corrected the error in Figure.
- Moreover, the descriptions in this figure are unreadable. Figure 5. The descriptions in this figure are unreadable too.
Response 5: Figure 4 and Figure 5
Thank you very much for your valuable advice. We have replaced the Figures 4 and 5 with high-definition pictures.
- Figure 8 and Figure 9. The name of coatings are missed. I propose to introduce the designations of the tested samples in section "Materials and methods" and use these designations throughout the manuscript.
Response 6: lines 101 to 105
Thank you very much for your valuable suggestions. We have added the name of coatings in Figures 8 and 9.Meanwhile, we have introduced the designations of the tested samples in section "Materials and methods" from lines 101 to 105.
7.Figure 10. The title of this figure is incorrect. The images present the surface morphology with wear traces (!). Please change it.
Response 7:
Thank you very much for pointing out the mistake on the title of Figure 10. We have corrected it to ‘the surface morphology with wear traces of coatings’ After our consideration, the content of Figure 10 seems to be repeated. Please allow us to delete Figure 10. Thank you very much for your correction.
Special thanks to you for your good comments.

Round 2
Reviewer 2 Report
The authors have effectively addressed most of my comments from the previous version, and I'm glad that these comments helped them correct errors in data analysis. There is one part of the revised manuscript that could still be clearer:
The added formula and discussion describing the growth rate in relation to dendrites is helpful to clarify earlier issues, but raises additional questions. Frist, is the product of nu and rho^2 always constant regardless of field strength? Second, what does it mean “When the EFS was exceed”? Are you referring to the case when EFS>20 mT, as in Figure 4c? It would be nice to see some relation between this new Eq. 1 and the actual growth rates under the three conditions reported.
Author Response
Dear Reviewer
On behalf of my co-authors, we thank you very much for giving us an opportunity to revise our manuscript, we appreciate you very much for the positive and constructive comments and suggestions on our manuscript entitled “Effect of electromagnetic field on wear resistance of Fe901/Al2O3 metal matrix composite coating prepared by laser cladding”. (Manuscript ID: materials-1588007).
We have studied reviewer’s comments carefully and have made revision which marked in red in the paper. We have tried our best to revise our manuscript according to the comments. Attached please find the revised version, which we would like to submit for your kind consideration.
We would like to express our great appreciation to you and reviewers for comments on our paper. Looking forward to hearing from you.
Thank you and best regards.
Sincerely yours,
Yaobang Chen
Responds to the reviewer’s comments:
Comments and Suggestions for Authors
The authors have effectively addressed most of my comments from the previous version, and I'm glad that these comments helped them correct errors in data analysis. There is one part of the revised manuscript that could still be clearer:
1.The added formula and discussion describing the growth rate in relation to dendrites is helpful to clarify earlier issues, but raises additional questions. Frist, is the product of nu and rho^2 always constant regardless of field strength? Second, what does it mean “When the EFS was exceed”? Are you referring to the case when EFS>20 mT, as in Figure 4c? It would be nice to see some relation between this new Eq. 1 and the actual growth rates under the three conditions reported.
Response 1:
Thank you very much for your valuable comments.
According to the reference (Journal of Materials Research 26, 1688-1695 (2011)), the product of and was constant during the grain growth regardless of the strength of the electromagnetic field. As you said, "when the EFS exceed" referring to 40 mT in the article, and we have modified in the text. Through the comparison of Fig. 4 (a1-c1), it is found that in Fig. 4 (a1) ρ is the Maximum, according to Formula 1, the actual growth rate is the smallest; Fig. 4 (b1) medium ρ is the Minimum, indicating the maximum actual growth rate.

Reviewer 3 Report
I have no more comments.
Author Response
Dear Reviewer
On behalf of my co-authors, we thank you very much for giving us an opportunity to revise our manuscript, we appreciate you very much for the positive and constructive comments and suggestions on our manuscript entitled “Effect of electromagnetic field on wear resistance of Fe901/Al2O3 metal matrix composite coating prepared by laser cladding”. (Manuscript ID: materials-1588007).
We would like to express our great appreciation to you and reviewers for comments on our paper.
Thank you and best regards.
Sincerely yours,
Yaobang Chen
